# Validating the Benefit of Combining Imaging and Clinical Data for Ischemic Stroke Outcome Prediction

**Zeyad Abouyoussef**[1,2]                    ZEYAD.ABOUYOUSSEF@UCALGARY.CA
**Johanna Ospel**[2,3]                          JOHANNA.OSPEL@UCALGARY.CA
**Roberto Souza**[2,4]                          ROBERTO.SOUZA2@UCALGARY.CA

[1] *Biomedical Engineering, University of Calgary, Calgary, Alberta, Canada*

[2] *Hotchkiss Brain Institute, University of Calgary, Calgary, Alberta, Canada*

[3] *Department of Diagnostic Imaging and Clinical Neurosciences, University of Calgary, Calgary, Alberta, Canada*

[4] *Electrical and Software Engineering, University of Calgary, Calgary, Alberta, Canada*

**Editors:** Accepted for publication at MIDL 2026

## Abstract

Endovascular treatment (EVT) has been proven to be a successful treatment for some cases of acute ischemic stroke. However, neuro-radiologists rely on a small set of clinical features to select patients for treatment. This leads to the exclusion of patients who would have benefited from treatment and the inclusion of others who would not have benefited from it. Deep learning has been used to predict stroke outcome from baseline imaging and clinical data, with most studies reporting that combining imaging and clinical data slightly outperforms classical methods (e.g., logistic regression) trained on clinical data only. However, it is not clear how much of this improvement is attributed to the imaging data and whether it is robust to larger and more diverse test sets. We use one of the largest multi-center acute ischemic stroke datasets ($n = 1,105$) to determine whether combining imaging and clinical data outperforms classical methods. We show that combining imaging and clinical data matches the performance of logistic regression (0.72 Area Under Receiver Operating Characteristic Curve (AUROC)) when evaluated on a multi-center test set of over 600 samples. We examine the models' predictions and weights and find that 1) both methods match each other's prediction for 78% of the samples, and 2) the weights associated with the imaging features are small compared to the clinical ones. This suggests that imaging features extracted from the deep learning model do not contribute to the prediction as much as the clinical ones.

**Keywords:** ischemic stroke, deep learning, outcome prediction, multimodal learning.

## 1. Introduction

Acute stroke is a leading cause of disability and death, with more than three million deaths worldwide (Martin et al., 2024). Acute ischemic stroke develops when a blood clot blocks one of the major arteries, causing brain tissue death (Martin et al., 2024). For acute ischemic stroke patients, time is of the essence. For every minute the patient is left untreated, approximately 2 million neurons die, leading to irreversible brain damage and a poor long-term prognosis (Saver, 2006). This risk requires fast decision-making to begin treatment as soon as possible and prevent long-term disabilities.

Endovascular treatment (EVT) is considered the most effective treatment for acute ischemic stroke, with multiple studies reporting improvement in patient outcome described by

the 90-day modified Rankin Scale (mRS) (Berkhemer et al., 2015; Goyal et al., 2015; Jovin et al., 2015). The mRS measures the patient's degree of disability while performing daily activities on a 7-point scale ranging from 0 (no symptoms) to 6 (death) where a good outcome is defined as mRS $\leq 2$ (Hill et al., 2020; Saver et al., 2021). Clinicians rely on features like the Alberta Stroke Program Early Computed Tomography Score (ASPECTS), clot location, and stroke onset time to select patients for EVT (Powers et al., 2019). However, it is challenging to select patients who will benefit from the procedure using such a small subset of clinical features. Furthermore, radiological features like ASPECTS do not fully capture the entire imaging information, ignore infarct morphology, and fail to accurately predict the 90-day mRS (Barber et al., 2000; Gupta et al., 2012; Farzin et al., 2016).

Multiple studies have used classical machine learning models (e.g., logistic regression) to estimate the mRS from clinical variables (Nishi et al., 2019; Venema et al., 2021; Rajashekar et al., 2021). However, only a small number of imaging-derived features (e.g., ASPECTS and clot location) were used to train the models. While these features are useful, they lack the full imaging information present in the scans. On the other hand, multiple studies have used deep learning to automatically learn useful features from the scans and estimate the mRS (Hilbert et al., 2019; Nishi et al., 2020; Samak et al., 2020; Bacchi et al., 2020; Samak et al., 2022; Jo et al., 2023; Samak et al., 2023; Amador et al., 2025; Liu et al., 2025; Diprose et al., 2025). When trained solely on medical images, deep learning models underperformed when compared to classical models. Only when jointly trained on clinical and imaging data do the models start to marginally outperform logistic regression, albeit on small test sets. So far, it is unclear why deep learning models only start performing once clinical data is added, and whether the improvement over classical methods is robust when tested on larger multi-center test sets.

Ramos et al. conducted several experiments on a cohort of $\sim 3,000$ patients, and their cross-validation results showed that combining imaging and clinical data underperformed logistic regression by 4% (Ramos et al., 2022). While they only trained a small ResNet model (He et al., 2016) on CT Angiography (CTA) scans, their findings imply that adding clinical data does not help the model learn useful imaging features. Instead, we believe it encourages the model to partially ignore imaging features and focus more on clinical features.

In this study, we explore the effect of combining images with clinical data for stroke outcome prediction. We start by training models that either use clinical or imaging data. Then, we explore the effect of combining them and show that models trained with both perform similarly to classical models. We examine model weights and explainability maps and show that clinical data have a bigger impact on the model's predictions than imaging data. Finally, we briefly investigate whether imaging-derived features are important to classical models.

## 2. Methods

### 2.1. Dataset

We used the ESCAPE-NA1 trial (Hill et al., 2020) dataset to train and evaluate our models. It contains pre-treatment non-contrast CT (NCCT), CTA, and clinical data for acute ischemic stroke patients undergoing EVT, as well as the 90-day mRS. Data from $1,105$

patients was collected from 48 acute care centers spanning eight countries. The trial investigators did not specify a specific acquisition protocol for each scan, and it was left to each center to use its standard protocol. This provides us with a diverse imaging dataset that contains a variety of scanners and acquisition parameters. The dataset contained $1,051$ samples with full baseline clinical and imaging data. We dichotomized the 90-day mRS such that a good outcome was defined as mRS $\leq 2$, and a bad outcome was defined as mRS $\geq 3$. Table 1 shows a summary of the variables in the dataset.

We split the data into training and testing splits. Unlike other studies (Samak et al., 2020, 2022, 2023) that pooled all samples from all care centers and randomly split, we grouped samples by care center and assigned all samples from each center to either training or testing. While assigning centers to the training set, we ensured that the final training set was balanced. This eliminates biases introduced from training on imbalanced data and testing on care centers already used during training. In total, our training set contained 400 samples from 18 care centers, while the test set contained 651 samples from 30 care centers. Finally, we performed 5-fold cross-validation on the training split for hyperparameter tuning. While this splitting scheme uses only about 40% of the available data for training, we generate a large test split that is unbiased to any specific center or acquisition protocol used in training. Furthermore, the amount of data used for training is similar to other studies (Samak et al., 2020, 2022, 2023).

## 2.2. Preprocessing

### 2.2.1. Imaging Data

We used TotalSegmentator (Wasserthal et al., 2023; Isensee et al., 2020) to automatically skull-strip the NCCT and CTA scans. Then, we performed foreground cropping to remove background regions. The NCCT scans were adjusted to a window level of 40 Hounsfield Units (HU) and a window width of 80 HU to highlight infarcted tissue, while the CTA scans were adjusted to a 250 HU window level and a 400 HU window width to highlight blood vessels. To account for variation in the training set, all NCCT scans were resampled to the training set's median voxel spacing of $0.45 \times 0.45 \times 2.5$ $mm^3$ and resized to a median image shape of $291 \times 356 \times 54$. As for the CTA scans, the median voxel spacing was $0.47 \times 0.47 \times 0.63$ $mm^3$, and the median image shape after resampling was $278 \times 337 \times 221$ $mm^3$. Voxel intensity was normalized based on the global mean ($\mu_{NCCT} = 29.81, \mu_{CTA} = 43.53$) and standard deviation ($\sigma_{NCCT} = 12.06, \sigma_{CTA} = 17.64$) of the foreground voxels in the training set. Finally, we performed random flipping on the fly during training time to reduce overfitting.

### 2.2.2. Model Training

We start by training logistic regression, random forest, and neural network models to estimate the 90-day mRS from 20 baseline clinical features. Features included were: age, sex, baseline mRS, baseline National Institutes of Health Stroke Scale (NIHSS), baseline ASPECTS, time between stroke onset and treatment, location of vessel occlusion, affected brain hemisphere, and medical history (i.e., smoking, diabetes, hypertension, atrial fibrillation, peripheral vascular disease, chronic renal failure, ischemic heart disease, congestive heart failure, high cholestrol, past stroke, recent stroke, major surgery). The location of vessel occlusion variable was one-hot encoded as it had 5 different values (i.e., distal M1

Table 1: Summary of the dataset variables with respect to the 90-day mRS. Continuous variables are represented by the median and interquartile range, while categorical variables are represented by percentages. mRS, modified Rankin Scale; ASPECTS, Alberta Stroke Program Early Computed Tomography Score; NIHSS, National Institutes of Health Stroke Scale; MCA, Middle Cerebral Artery.

| Variable | All samples ($n = 1051$) | mRS $\leq 2$ ($n = 640$) | mRS $\geq 3$ ($n = 411$) |
|---|---|---|---|
| Age | 70.8 (60.7-79.7) | 66.4 (57.98-75.83) | 76.65 (67.53-83.57) |
| Sex, Male | 50.38% | 53.08% | 46.91% |
| Time between stroke onset and treatment | 201 (135-326.75) | 185 (127-286) | 230 (148.5-369) |
| ASPECTS | 8 (7-9) | 8 (7-9) | 8 (7-9) |
| NIHSS | 17 (12-21) | 16 (11-20) | 18 (14-22) |
| Hemishphere Stroke, Left | 47.26% | 45.20% | 50.47% |
| Occlusion Site, MCA | 80.23% | 84.11% | 74.15% |
| Hypertension | 69.87% | 65.43% | 76.81% |
| Smoker | 49.15% | 52.31% | 47.69% |
| Diabetes | 19.68% | 14.66% | 27.53% |
| Atrial Fibrillation | 35.03% | 29.63% | 43.48% |
| Peripheral Vascular Disease | 5.37% | 4.32% | 7.00% |
| High Cholestrol | 46.67% | 43.05% | 52.42% |
| Chronic Renal Failure | 5.46% | 4.17% | 7.49% |
| Chronic Heart Failure | 12.24% | 11.57% | 13.29% |
| Recent Major Surgery | 3.58% | 3.70% | 3.38% |
| Past Stroke | 13.84% | 12.96% | 15.22% |
| Recent Stroke | 4.50% | 4.70% | 4.20% |
| Ischemic Heart Disease | 22.88% | 20.37% | 26.81% |

Middle Cerebral Artery (MCA), proximal M1 MCA, mid M1 MCA, M2/M3 MCA, Internal Carotid Artery (ICA)). The logistic regression model was trained for up to $5,000$ iterations with $l2$ regularization. The random forest model had $1,000$ estimators, with a maximum tree depth of 10, a minimum of 2 samples at leaf nodes, a minimum of 5 samples to split an internal node, and a maximum of 4 features considered for the best split. The neural network consisted of a linear layer with 512 output features, a batch normalization layer, a linear layer with 512 features, a ReLU, and a final classification layer.

We chose a 3D ResNeXt (Xie et al., 2017) with Squeeze-and-Excitation (SE) (Hu et al., 2018) modules as the backbone for extracting features from the scans. ResNeXt extends ResNet by using grouped convolutions to make the network more efficient. This reduces the high computational cost associated with training on 3D data, leading to faster convergence. Meanwhile, SE acts as a channel attention mechanism that helps the network capture the most important channels in a feature map. In total, the backbone had 29,401,840 parameters with the final layer outputting 512 features. The 512 features are batch normalized, then fed to a classification head consisting of a linear layer, a ReLU, and a final linear layer that predicts the mRS.

When combining NCCT with CTA, we use separate backbones for each scan type and concatenate the features after batch normalization. The concatenated features are then passed to the classification head for mRS prediction. Combining clinical data with one scan type follows a similar procedure. We feed the clinical data into a linear layer with 512 output features and concatenate them with the imaging ones. When combining all data, we feed each input to its own backbone, concatenate, and then proceed to the classification head. Figure 1 shows how we combine features in different training setups.

In addition to training with the ResNeXt backbone, we train with two different imaging backbones when combining NCCT and clinical data. First, we retrain the TranSOP model by Samak et al. (Samak et al., 2023). Their model had a similar setup to ours, but replaces the ResNeXt backbone with a Swin transformer (Liu et al., 2021). Second, we use the pretrained Med3D network as an imaging backbone (Chen et al., 2019). Med3D was originally trained on 8 datasets containing scans and segmentation maps of different tumors and organs.

In all experiments, we pick the fold with the highest F1 score and use it for evaluation on the test set. Models were developed using MONAI (Cardoso et al., 2022) and scikit-learn (Pedregosa et al., 2011) and trained on 80 GB A100 and H100 NVIDIA GPUs. Code is publicly available at https://github.com/zeyad-kay/multimodal_mrs_prediction

## 3. Results

In total, we trained 11 models to investigate how combining clinical and imaging data affects model performance. Table 2 summarizes the performance of each model along with uncertainty estimates for each metric. Logistic regression was the best clinical-only model with 0.72 Area Under Receiver Operating Characteristic Curve (AUROC). Figure 2 shows the SHapley Additive exPlanations (SHAP) values (Lundberg and Lee, 2017) for the logistic regression model. Features like age, ASPECTS, NIHSS, and time between stroke onset and treatment contributed the most to the model predictions.

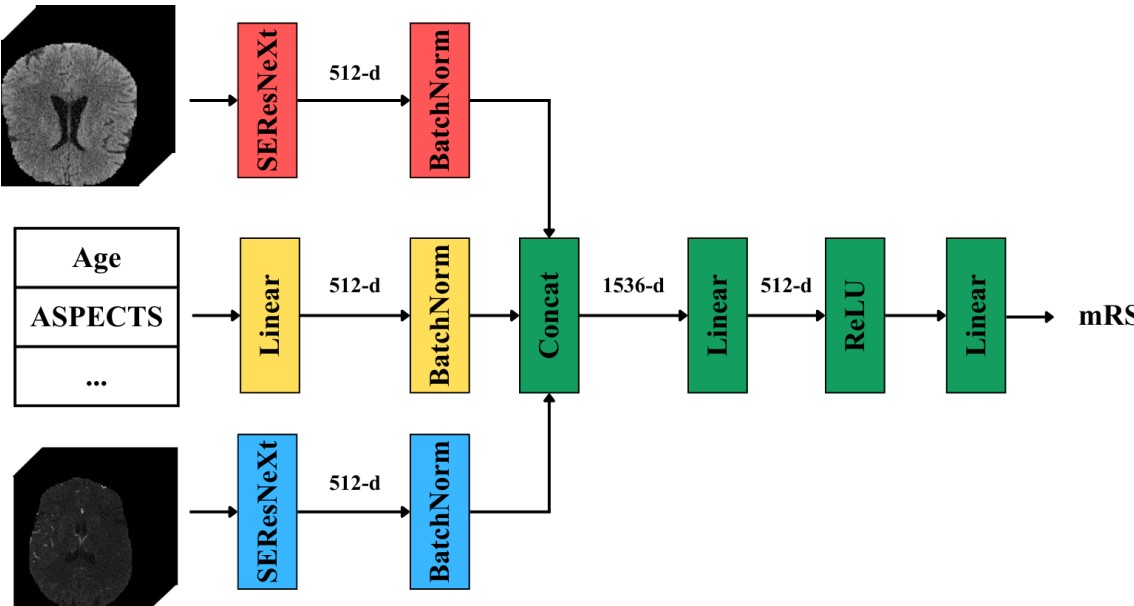

Figure 1: Network architecture used when training the models. Each input is fed into a separate backbone (red, yellow, blue) that extracts a 512-dimensional feature vector. The features are concatenated and fed to a linear and a final classification layer (green). Backbones are enabled/disabled depending on which input data we are using. For example, we only use the red backbone when training on NCCT, the yellow and blue when training on clinical data and CTA, and all backbones when training on NCCT, CTA, and clinical data. SEResNeXt, Squeeze-and-Excitation ResNeXt.

Training with only the scans resulted in lower AUROC than the clinical-only models. The NCCT model was the best imaging-only model, followed by the NCCT + CTA combination. Combining imaging and clinical data outperformed the imaging-only models, with the NCCT + Clinical model matching the AUROC of logistic regression and achieving the best Area Under Precision Recall Curve (AUPRC) across all experiments. Meanwhile, the TranSOP model achieved the best F1 score and accuracy. While the CTA-only and NCCT + CTA experiments had high F1 scores, they had 96% and 85% false positive rates, respectively.

## 4. Discussion

The reduced performance of the imaging-only models compared to the ones using clinical data is consistent with current literature (Samak et al., 2020, 2022, 2023; Liu et al., 2025). For instance, Samak et al. reported that training with only clinical data outperformed training with NCCT by 6% AUROC (0.73 vs 0.67) (Samak et al., 2023). While Hilbert et al. reported better AUROC when training with 2D maximum intensity projections of

Table 2: Test set performance across different models. Best model highlighted in bold. A random model has 0.67 AUPRC and 0.5 AUROC. Model labeled "All" is the NCCT + CTA + Clinical. CI, Confidence Interval; AUROC, Area Under Receiver Operating Characteristic Curve; AUPRC Area Under Precision Recall Curve; LR, Logistic Regression; RF, Random Forest; NN, Neural Network.

| Models | AUROC (95% CI) | AUPRC (95% CI) | F1 (95% CI) | Accuracy (95% CI) |
|---|---|---|---|---|
| **Clinical** | | | | |
| LR | **0.72 (0.68-0.77)** | 0.83 (0.79-0.87) | 0.75 (0.71-0.78) | 0.68 (0.64-0.72) |
| RF | 0.7 (0.66-0.74) | 0.81 (0.77-0.85) | 0.75 (0.71-0.78) | 0.68 (0.64-0.71) |
| NN | 0.65 (0.6-0.69) | 0.77 (0.72-0.81) | 0.68 (0.65-0.72) | 0.61 (0.58-0.65) |
| **Imaging** | | | | |
| NCCT | 0.57 (0.52-0.62) | 0.72 (0.66-0.77) | 0.51 (0.47-0.56) | 0.49 (0.45-0.53) |
| CTA | 0.5 (0.45-0.55) | 0.66 (0.61-0.71) | 0.78 (0.76-0.81) | 0.65 (0.61-0.69) |
| NCCT+CTA | 0.55 (0.5-0.59) | 0.71 (0.66-0.76) | 0.77 (0.74-0.8) | 0.65 (0.61-0.68) |
| **Imaging+Clinical** | | | | |
| NCCT+Clinical | **0.72 (0.68-0.76)** | **0.84 (0.8-0.87)** | 0.75 (0.72-0.79) | 0.68 (0.65-0.72) |
| TranSOP | 0.71 (0.66-0.75) | 0.82 (0.77-0.86) | **0.79 (0.76-0.82)** | **0.7 (0.67-0.74)** |
| Med3D | 0.71 (0.67-0.75) | 0.83 (0.79-0.87) | 0.75 (0.72-0.78) | 0.68 (0.64-0.71) |
| CTA+Clinical | 0.62 (0.58-0.67) | 0.76 (0.72-0.8) | 0.77 (0.75-0.8) | 0.66 (0.62-0.69) |
| All | 0.66 (0.61-0.7) | 0.8 (0.75-0.83) | 0.67 (0.63-0.71) | 0.6 (0.56-0.64) |

CTA scans (0.71 vs 0.68), their results were based on internal cross-validation data (Hilbert et al., 2019). Interestingly, our results show that training with CTA performed worse than with NCCT, even when combined with the clinical data. This might be attributed to the HU window highlighting blood vessels as opposed to the one highlighting infarcts in NCCT. This assumption is based on several factors: 1) SHapley Additive exPlanations values (Lundberg and Lee, 2017) in Figure 2 show that baseline ASPECTS tends to have a larger impact on model predictions than occlusion location, 2) Diprose et al. (Diprose et al., 2025) reported that combining both imaging modalities yielded better performance, but their analysis showed that they are using a similar window for both modalities, one that highlights infarcts. To validate this assumption, we retrained the CTA-only, CTA + Clinical, and NCCT + CTA + Clinical models with the same window used in NCCT. The retrained CTA-only model improved AUROC to 0.56 and AUPRC to 0.72, the CTA + Clinical model improved AUROC to 0.69 and AUPRC to 0.79, and the NCCT + CTA + Clinical model improved AUROC to 0.69. This confirms that the wide window highlighting blood vessels is not as helpful to the model as the one highlighting infarcts.

The superior performance of the models combining imaging and clinical data over the imaging-only models is indicative of the importance of clinical data to the models. Again, several studies highlight this pattern (Samak et al., 2020, 2022; Ramos et al., 2022; Samak et al., 2023; Liu et al., 2025). Samak et al. combined baseline NCCT and clinical data to achieve a state-of-the-art performance of 0.85 AUROC (Samak et al., 2023). While their

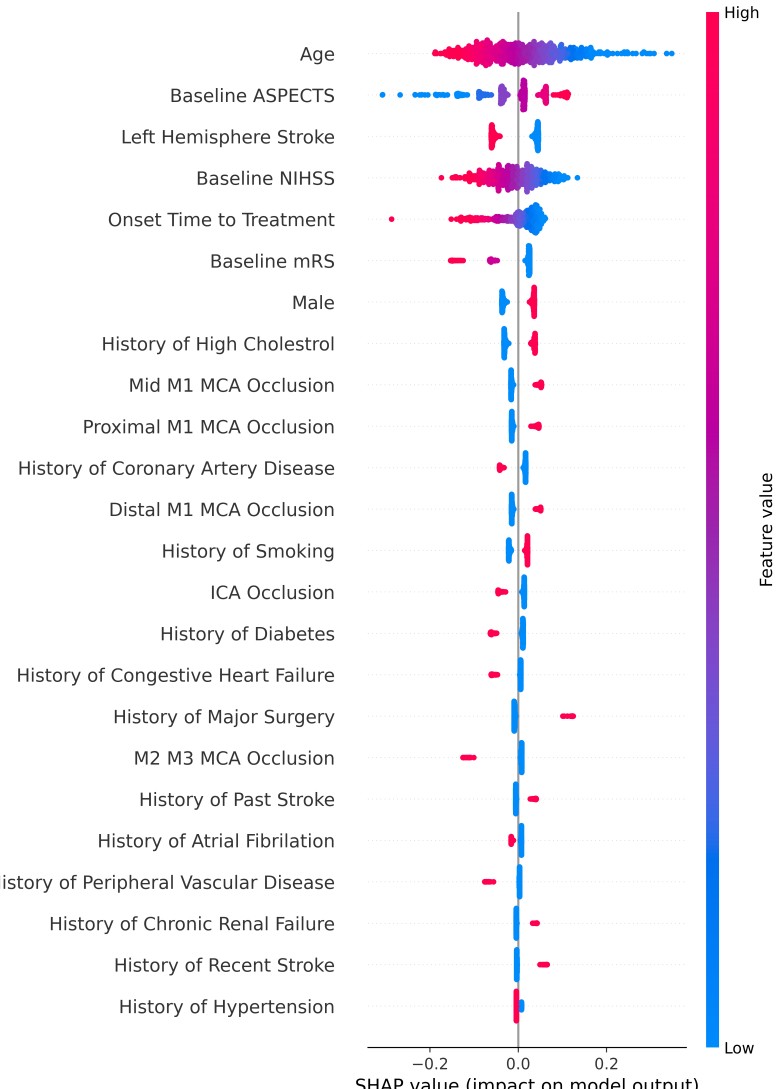

Figure 2: SHapley Additive exPlanations (SHAP) values for the logistic regression model. Positive SHAP values are associated with good outcomes, while negative ones are associated with bad outcomes. Features are ordered based on their importance, with the most important ones being at the top. High feature values are depicted in red while low ones are in blue. For example, older patients (red) are associated with bad outcomes (negative SHAP value).

model showed a 20% improvement over training with only NCCT, they only tested on a cohort of 75 patients. To investigate how changing the backbone affects performance, we retrained their model. On our test set, the model's AUROC dropped to 0.71. This indicates

that even when changing the imaging backbone to a more advanced architecture, it did not outperform logistic regression.

It is worth noting that having an independent test set from centers not used in training is beneficial for evaluating the models on unseen data. However, the reduced performance of our models might be due to a large distribution shift between the training and test sets. To test this, we pooled all patients from all centers and randomly split the data. These results are summarized in Table 3 in the appendix.

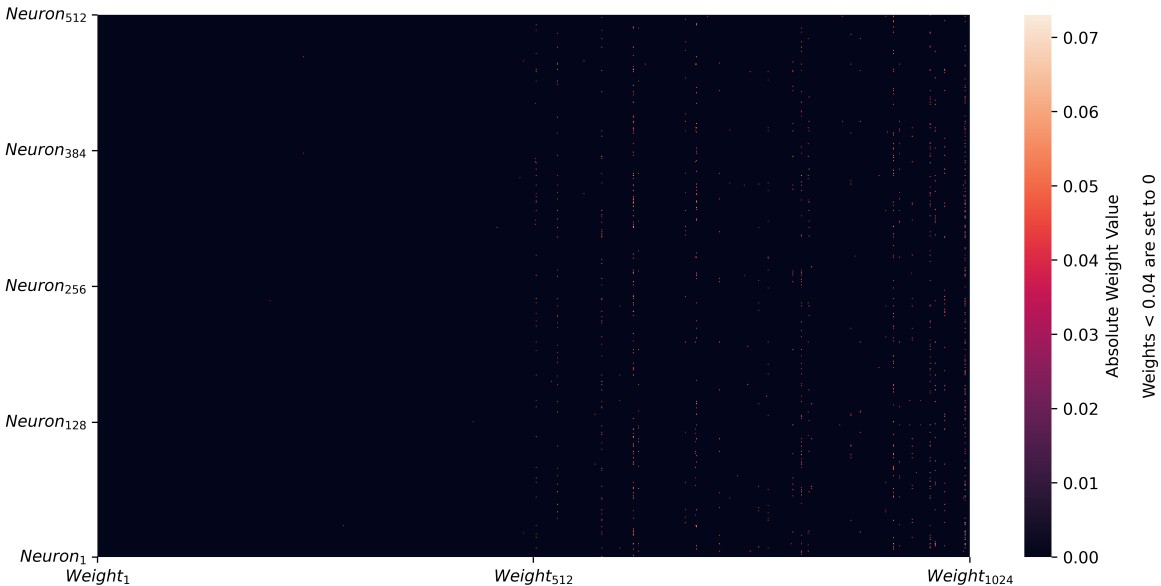

Figure 3: Heatmap of the weights of the linear layer following feature concatenation in the NCCT + Clinical model. The layer has 512 neurons (y-axis) and 1024 weights for each neuron (x-axis). The first 512 weights (left half) are the ones associated with the imaging features extracted from SEResNeXt, and the second 512 weights (right half) are the ones associated with the clinical features. Weights with absolute value $< 0.04$ are set to zero for better visualization.

The near identical performance between logistic regression and NCCT + Clinical is interesting. When looking at the predictions from both models, we find that the predictions are identical for 78% of the test samples. This might imply that the NCCT + Clinical model is just focusing on the clinical features. While it is challenging to fully determine how much the network learns from the imaging data, we tackle this question by 1) examining the weights of the linear layer following feature concatenation, and 2) estimating SHAP values for the inputs. Figure 3 shows the weights at each neuron in the linear layer following feature concatenation of the NCCT + Clinical model. Each neuron has a 1024-dimensional weight vector $w$ with $w_{1,...,512}$ being associated with imaging features and $w_{513,...,1024}$ with the clinical features. When we clip small weights with absolute value $< 0.04$ to zero, we see that almost all imaging weights across all neurons are clipped, while the clinical weights

have larger absolute values and were not clipped. A similar pattern is also observed in the Med3D model as shown in Figure 6 in the appendix.

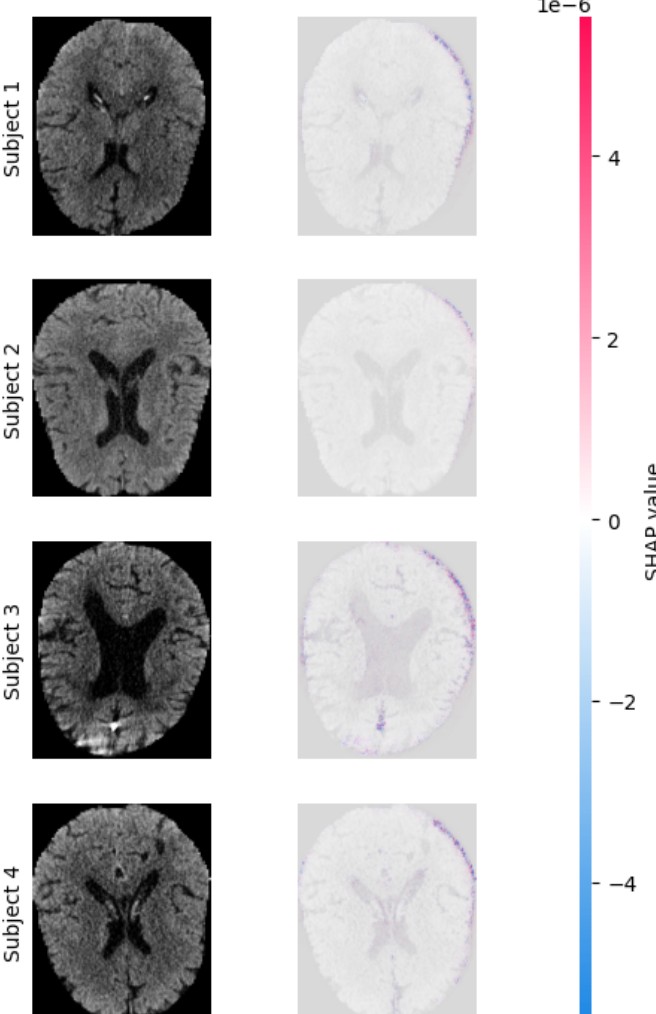

Figure 4: Explainability maps of 4 different subjects for the NCCT + Clinical model. Each row contains an NCCT slice of a different subject along with overlayed SHAP values. Pixels with positive SHAP values (red) influence the model to predict a good outcome, while negative SHAP values (blue) influence the model to predict a bad outcome. Subject 1 was correctly classified as having a good outcome. Subject 2 was incorrectly classified as having a good outcome. Subject 3 was correctly classified as having a bad outcome. Subject 4 was incorrectly classified as having a bad outcome.

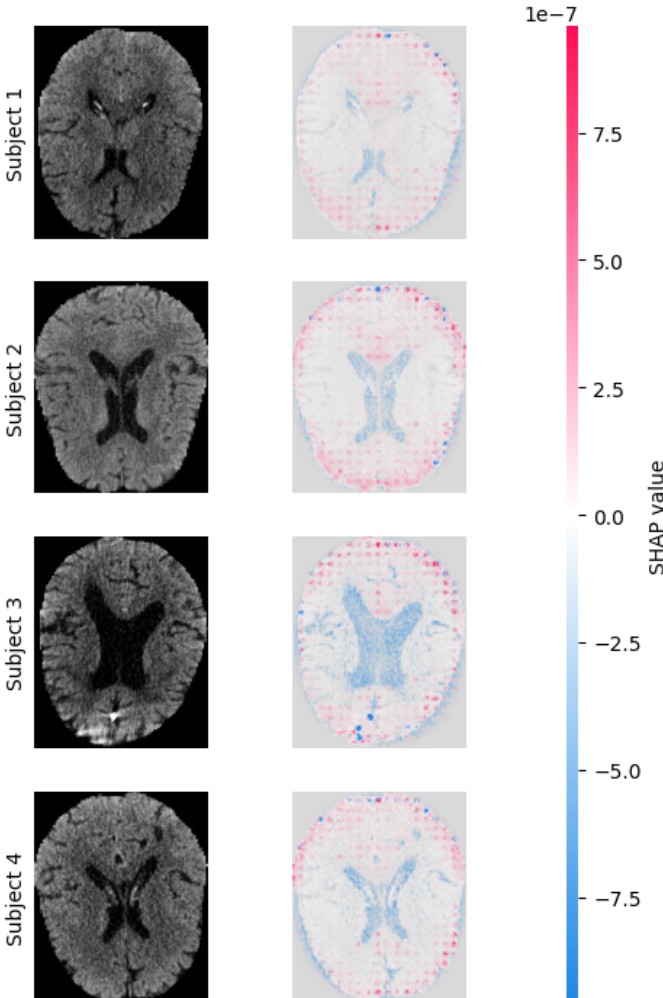

Figure 5: Explainability maps of 4 different subjects for the Med3D model. Each row contains an NCCT slice of a different subject along with overlayed SHAP values. Pixels with positive SHAP values (red) influence the model to predict a good outcome, while negative SHAP values (blue) influence the model to predict a bad outcome. Subject 1 was correctly classified as having a good outcome. Subject 2 was incorrectly classified as having a good outcome. Subject 3 was correctly classified as having a bad outcome. Subject 4 was incorrectly classified as having a bad outcome.

As for the SHAP values, Figure 4 shows the explainability maps of 4 different subjects for the NCCT + Clinical model. Clearly, the model does not focus on relevant parts of the brain. Since each scan contains millions of pixels, the contribution of each pixel to the prediction is relatively small. Hence, we sum SHAP values of all pixels across all

slices to get the net contribution of the scan. Then, we compare this value to the sum of SHAP values associated with the input clinical features. This helps us directly compare the contribution of each modality. Out of 17 patients we have analyzed, 12 had close to zero NCCT net contribution. While this might be confounded by the fact that the imaging backbone failed to learn useful image representations due to being trained from scratch, the SHAP values for the Med3D model shown in Figure 5 still had close to zero NCCT net contribution. Although this is a small patient sample, it supports what we have already observed in Figure 3. Additional explainability maps of the NCCT-only model are presented in Figure 7 in the appendix.

Surprisingly, the multimodal model does not seem to be benefiting from the imaging features, especially since Figure 2 shows that ASPECTS, an imaging-derived feature, is important for logistic regression. To further investigate this, we retrain the logistic regression model without any imaging-derived information (i.e., ASPECTS and occlusion location) and find that the AUROC and accuracy stay the same, AUPRC increases by 1%, and $F1$ decreases by 1%. This indicates that even when training with logistic regression, imaging-derived features like ASPECTS do not affect performance. This is surprising since ASPECTS is considered a proxy to the total infarct volume, a variable shown by Ospel et al. to have a strong association with good outcomes (Ospel et al., 2024).

While we show that imaging features provide limited value to the models, our work has several limitations. The absence of lesion segmentation masks makes it harder for the imaging backbones to learn meaningful representations. While we do have follow-up lesion volume estimates, pretraining the imaging backbones to predict lesion volume failed despite trying different hyperparameter configurations. Another limitation is using a simple concatenation strategy for combining imaging and clinical data. While this strategy is used by relevant literature and is easier for visualizing model weights, it might have hindered the model's ability to learn better image representations.

## 5. Conclusion

In this study, we investigated how different inputs affect stroke outcome prediction model performance. We found that training with only imaging data drastically reduces performance, while combining imaging and clinical data results in identical performance to logistic regression. Finally, we examined the multimodal model weights and found that imaging features do not contribute to the prediction as much as the clinical features. Future work will focus on improving the learned image representations by training more advanced architectures and investigating how other fusion strategies like cross attention affect performance.

## Acknowledgments

The study was funded by an NSERC Discovery Grant awarded to Roberto Souza (#RGPIN-2021-02867) and is supported by the Hotchkiss Brain Institute. The authors also thank the Digital Research Alliance of Canada and the University of Calgary Research Computing Services for providing access to GPU clusters.

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

# Appendix A. Random Splitting Strategy

To investigate whether our splitting strategy of using a test set that does not contain centers used in training affected model performance due to distribution shift, we pool all patients from all centers and randomly split the data. Table 3 shows the results of model performance.

Table 3: Test set performance across different models when using a random splitting strategy. In this splitting strategy, centers can appear in the training and testing sets. Best model highlighted in bold. A random model has 0.67 AUPRC and 0.5 AUROC. CI, Confidence Interval; AUROC, Area Under Receiver Operating Characteristic Curve; AUPRC Area Under Precision Recall Curve; LR, Logistic Regression; RF, Random Forest; NN, Neural Network.

| Models | AUROC (95% CI) | AUPRC (95% CI) | F1 (95% CI) | Accuracy (95% CI) |
|---|---|---|---|---|
| **Clinical** | | | | |
| LR | **0.75 (0.71-0.79)** | **0.85 (0.81-0.88)** | 0.74 (0.71-0.77) | **0.68 (0.65-0.72)** |
| RF | 0.72 (0.68-0.76) | 0.83 (0.79-0.87) | 0.71 (0.68-0.75) | 0.65 (0.61-0.69) |
| NN | 0.69 (0.65-0.74) | 0.81 (0.77-0.85) | 0.72 (0.68-0.75) | 0.65 (0.62-0.69) |
| **Imaging** | | | | |
| NCCT | 0.61 (0.56-0.65) | 0.75 (0.71-0.8) | 0.75 (0.72-0.78) | 0.65 (0.61-0.68) |
| CTA | 0.47 (0.42-0.52) | 0.64 (0.6-0.69) | 0.81 (0.78-0.83) | 0.68 (0.65-0.71) |
| NCCT+CTA | 0.59 (0.55-0.64) | 0.72 (0.67-0.77) | 0.7 (0.66-0.73) | 0.6 (0.56-0.64) |
| **Imaging+Clinical** | | | | |
| NCCT+Clinical | 0.71 (0.67-0.75) | 0.83 (0.79-0.87) | 0.71 (0.68-0.75) | 0.65 (0.62-0.69) |
| TranSOP | 0.7 (0.66-0.73) | 0.81 (0.77-0.85) | 0.76 (0.73-0.79) | **0.68 (0.64-0.71)** |
| CTA+Clinical | 0.62 (0.57-0.66) | 0.77 (0.72-0.81) | 0.79 (0.77-0.82) | 0.68 (0.64-0.71) |
| All | 0.62 (0.58-0.67) | 0.76 (0.71-0.8) | **0.79 (0.77-0.82)** | **0.68 (0.65-0.71)** |

# Appendix B. Med3D Model Weights

We visualize the heatmaps for the Med3D model in Figure 6. Weights associated to the clinical data have a larger absolute value than the imaging ones

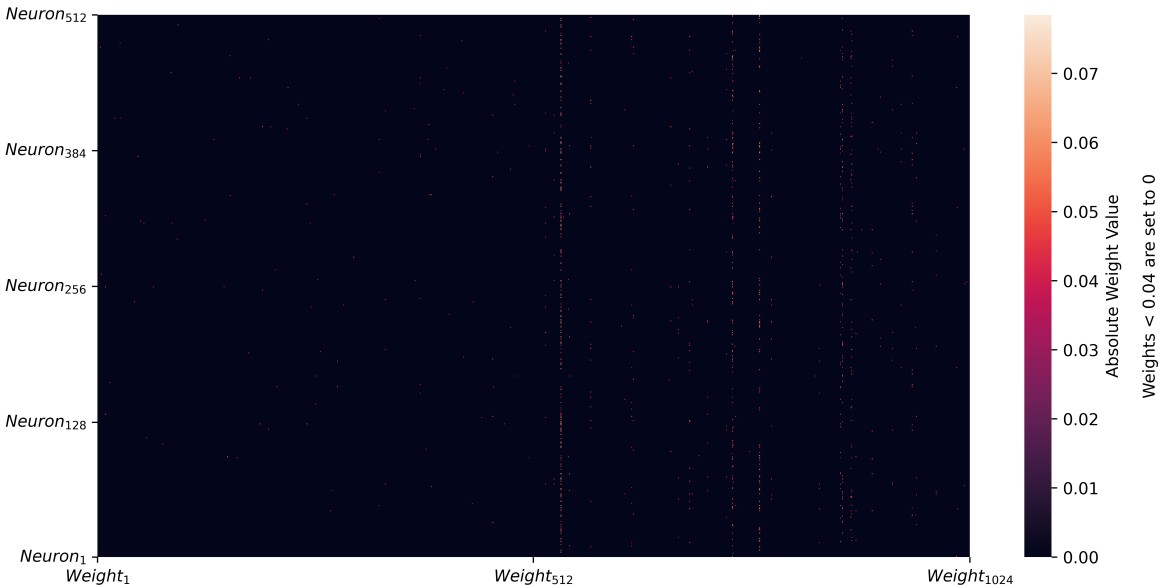

Figure 6: Heatmap of the weights of the linear layer following feature concatenation in the Med3D model. The layer has 512 neurons (y-axis) and 1024 weights for each neuron (x-axis). The first 512 weights (left half) are the ones associated with the imaging features extracted from Med3D, and the second 512 weights (right half) are the ones associated with the clinical features. Weights with absolute value < 0.04 are set to zero for better visualization.

## Appendix C. Explainability Maps

We estimate the explainability maps for the NCCT-only model in Figure 7. While the model seems to be focusing on different parts of the brain, the bad performance reported previously indicates that the model is not learning clinically meaningful features.

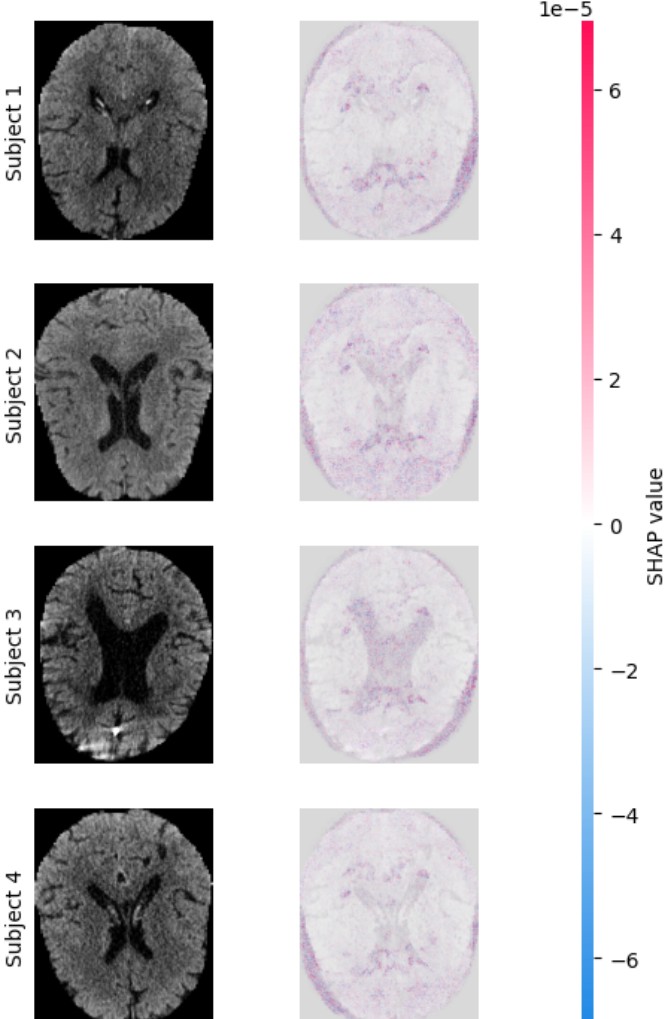

Figure 7: Explainability maps of 4 different subjects extracted from the NCCT-only model. Each row contains an NCCT slice of a subject along with overlayed SHAP values. Pixels with positive SHAP values (red) influence the model to predict a good outcome, while negative SHAP values (blue) influence the model to predict a bad outcome. Subjects 1 and 4 were incorrectly classified as having bad outcomes. Subject 2 was incorrectly classified as having a good outcome. Subject 3 was correctly classified as having a bad outcome.

