# OpenReview forum: "Validating the Benefit of Combining Imaging and Clinical Data for Ischemic Stroke Outcome Prediction"
_MIDL.io/2026/Conference — MIDL 2026 Poster_

### Official Review · Reviewer_fft7 · 2026-01-05

**Confidence:** 4
**Preliminary Rating:** 4
**Final Rating:** 4

**Summary:**

The paper introduce a novel method for predicting treatment related information for acute stroke patients from clinical and imaging data. The method presented employs a 3D ResNeXt to process contrast enhanced and non contrast computed tomography images and a concatenation operation for fusing clinical and imaging data. Experiments on a dataset of 1105 patients to compare different data modalities scenario. From the experimental results, the authors draw conclusions on the relative importance of the modalities for the prediction as well as the performance improvements. Further investigations brought nuances into certains conclusions while confirming others.

**Strengths:**

The paper is well written and contain most informations for reproducing the results (some clarifications are still needed though but a link to a github repo is provided). Experimental results confirm previously found insights of the literature and brings more questions regarding the importance of modalities for the prediction of the learning algorithms used. The metrics used correctly cover the conclusions drawn by the authors.

**Weaknesses:**

The paper lack a comparison with literature methods to better understand the contribution brought by the proposed architecture.
The authors did not clearly state how the target variable is important for the medical context.

**Detailed Comments:**

An interpretation analysis of the CNN attention maps (e.g., GradCAM) would provide more empirical insights into the decision of the models using imaging data.

In section 1, what does this statement means *"Furthermore, radiological features like ASPECTS[...] fail to fully describe the clinical outcome [...]."*?

**Justification Of Final Rating:**

The authors clearly answered all the comments made. They provided additional results confirming the one presented in the paper. Although we encourage the authors to include the results discussed in the review, the answers were satisfactory.

**Justification Of The Preliminary Rating:**

The preliminary rating is justified by the insights brought by the paper results and analysis, rather than the novelty of the method. The authors provide interesting conclusions on the role of modalities for the task and encourage an exploration to further understand the role of these modalities and why the model performance is relatively not impacted by the removal of certain clinical variables derived from images although their evaluated importance was high at first.

**Questions To Address In The Rebuttal:**

The authors did not clearly state how the target variable is important for the medical context: why is the mRS information important for selecting the patient to deliver the mentioned treatment.

On which basis (expertise) the dichotomization of the 90-day mRS was chosen?

In section 2.2.2 it is stated that clinical data include the baseline mRS, how is different from the target variables and is there any risk of target leakage in this information?

In table 2, TranSOP shows the best F1 score indicating that it has the best tradeoff capacity at identifying positive cases, shouldn't this be considered as the best method rather than AUROC or AUPRC? If no why?

---

> ### Author Response · Authors · 2026-01-24
> **Response**
>
> Thank you for the detailed feedback. Please see the answers below:
>
> **Comment**: An interpretation analysis of the CNN attention maps (e.g., GradCAM) would provide more empirical insights into the decision of the models using imaging data.
>
> **Response**: We fully agree with that comment. We performed Shap analysis on the deep learning models and found that 1) the input clinical features have a bigger contribution to the prediction and 2) the outer edges of the brain have large Shap values compared to the rest of the brain. We included our findings in the updated manuscript.
>
> **Comment**: In section 1, what does this statement means "Furthermore, radiological features like ASPECTS[...] fail to fully describe the clinical outcome [...]."?
>
> **Response**: It means that ASPECTS do not accurately predict mRS. We updated the manuscript to make it clearer for the reader.
>
> **Comment**: The authors did not clearly state how the target variable is important for the medical context: why is the mRS information important for selecting the patient to deliver the mentioned treatment.
>
> **Response**: When assessing the efficacy of a stroke treatment, stroke trial investigators use the 90-day mRS as the primary outcome (https://www.nejm.org/doi/full/10.1056/NEJMoa1411587,and https://www.thelancet.com/journals/lancet/article/PIIS0140-6736(20)30258-0/fulltext). We updated the introduction to emphasize that.
>
> **Comment**: On which basis (expertise) the dichotomization of the 90-day mRS was chosen?
>
> **Response**: We follow the same convention used by multiple stroke trials (https://www.nejm.org/doi/full/10.1056/NEJMoa1411587,and https://www.thelancet.com/journals/lancet/article/PIIS0140-6736(20)30258-0/fulltext). Furthermore, similar studies to ours that we have cited in the manuscript (e.g., https://www.sciencedirect.com/science/article/pii/S0010482519303786, and https://ieeexplore.ieee.org/document/10230576) follow the same convention when predicting mRS. We updated the manuscript to make it clearer.
>
> **Comment**: In section 2.2.2 it is stated that clinical data include the baseline mRS, how is different from the target variables and is there any risk of target leakage in this information?
>
> **Response**: There is no target leakage introduced by the baseline mRS because it is measured at hospital admission while our target variable is measured 90-days after treatment. Hence, the variable is used by clinicians to assess the patient's condition pre-treatment and is included in their decision-making process.
>
> **Comment**: In table 2, TranSOP shows the best F1 score indicating that it has the best tradeoff capacity at identifying positive cases, shouldn't this be considered as the best method rather than AUROC or AUPRC? If no why?
>
> **Response**: The F1 score is calculated at a specific threshold (0.5 in our case). While TranSOP has the best F1 score at 0.5 threshold, the other models might have a better score if we choose a different threshold. This is why we are using AUPRC instead of F1, we want to compare the performance of the models regardless of the threshold.

---

> > ### Comment · Reviewer_fft7 · 2026-01-27
> >
> > Thanks to the author for taking into consideration the review and updating the manuscript accordingly.
> >
> > I agree with the author regarding the image feature extractor, and looking at the number of training samples (<1000) and the size of the model (~30M params) this is clearly to be expected. A pre-trained backbone, especially trained on CT images, should probably have been used. It is very likely that the imaging model didn't learn much.

---

> > ### Author Response · Authors · 2026-01-27
> > **Response**
> >
> > We tried the Med3D pretrained image encoder backbone (https://arxiv.org/pdf/1904.00625) when training the NCCT + Clinical model. This backbone was originally trained on 8 datasets containing segmentations of different tumor types and organs. In a previous study that we have cited in our manuscript (https://doi.org/10.3389/FNEUR.2022.809343), they used the pretrained Med3D backbone to train on CTA and clinical features. While their results improved substantially when using it, they 1) only reported cross-validation results, and 2) the logistic regression model had 4% better AUROC.
> >
> > In experiments that we have not included in the manuscript, we trained the NCCT + Clinical model with the Med3D backbone and found that 1) it had similar performance (AUROC=0.71, AUPRC=0.83) to SEResNext and TranSOP, and 2) the weight heatmap showed a similar pattern to the one we show in the manuscript (i.e., larger weights for the clinical features). We plan on adding this experiment to the appendix following the recommendation of reviewer kpmd

---

> > > ### Comment · Reviewer_fft7 · 2026-01-27
> > >
> > > Thanks for the clarification. Indeed, this confirms what was found in the original experiments and is an interesting result to include.
> > >
> > > However, still such a model has too many parameters to be fitted in a such small dataset. I wouldn't expect it to perform well except if data augmentation is heavily used. Is it the case ?

---

> > > > ### Author Response · Authors · 2026-01-27
> > > > **Response**
> > > >
> > > > We train with different model sizes. When training with the pretrained backbone, the model has approx. 15M parameters. While the TranSOP one has approx. 11M parameters.

---

### Official Review · Reviewer_kpmd · 2026-01-08

**Confidence:** 4
**Preliminary Rating:** 3
**Final Rating:** 4

**Summary:**

This paper deals with critically validating whether combining imaging data with clinical data truly improves ischemic stroke outcome prediction, compared to traditional clinical-only models such as logistic regression or other simple machine learning methods. In particular, the authors aim to determine how much predictive value deep-learning based models provide by extracting the imaging features from say (NCCT,CTA images of stroke lesions) actually contribute when evaluated on a large, diverse, multi-center test set. The study has been conducted on a massive multi-centre data to analyse the above hypothesis.

**Strengths:**

The paper has several strengths. I'm particularly impressed by the large multi-centre study and also making sure that the train and validation splits contain samples from completely different centres which makes a robust evaluation strategy. I also think that this work highlights an important step in validation where it's important to systematically analyse whether multi-modal data integration actually contributes to improvement or not.

1. Clinically relevant and well-motivated question - The paper tackles an important problem setting of multi-modal combination ->  whether multimodal deep learning models truly add value over traditional clinical models for ischemic stroke outcome prediction. Given the growing enthusiasm for multimodal learning in medical imaging, a careful validation study that questions assumed benefits is both necessary and impactful. It's really important to understand this systematically.

2. Strong dataset and evaluation protocol - The use of the ESCAPE-NA1 dataset, with over 1000 patients from 48 centers, is a major strength. The decision to split data by care center rather than randomly is very well justified, as it better reflects real-world generalization and avoids subtle information leakage that may inflate performance in prior work.

3. Thoughtful experimental design and ablation strategy - The authors systematically evaluate clinical-only, imaging-only, and multimodal models, including a retraining of a recent SOTA multimodal transformer (TranSOP). The inclusion of multiple baselines (LR, RF, MLP) strengthens the experiment settings and benchmarking claims.

4. Interpretability-driven analysis - The authors don't only stop at comparing the accuracy, but also showing explainability and interpretability analysis in terms of which factors are actually important to the outcome prediction task - SHAP values for clinical models, and learned weight magnitudes in multimodal networks.

5. Clear negative result with practical implications - The finding that multimodal models largely ignore imaging features and that clinical-only LR matches or exceeds their performance on a large, heterogeneous test set is an important result that challenges prevailing assumptions in the field.

**Weaknesses:**

Although the problem statement is extremely important in this field and the overall study has been done systematically with a large dataset, there are many weakness which need clarification and confirmation on whether the reported results are actually correct. Please see the weakness below.

1. Image analysis methodology is not SOTA - While the use of a 3D SE-ResNeXt backbone is reasonable, it is not fully clear whether this architecture represents the strongest possible approach for extracting the best imaging features from stroke CT data. Especially given that the NCCT images are very low contrast and don't always provide the best representation of the stroke lesions, it becomes necessary to use a strong vision encoder backbone (for e.g., try DINOv3 or other equiv.) for extracting the imaging features. Recent works have also increasingly explored lesion-centric or region-of-interest–based representations, explicitly locating the infarct/penumbra by segmentation models and then extracting the features of only those areas (https://pmc.ncbi.nlm.nih.gov/articles/PMC11847528/, https://pmc.ncbi.nlm.nih.gov/articles/PMC11968395/). Moreover, I also think that some simple attention or cross-attention mechanisms that align imaging features with clinical variables should also have been tried for multimodal prediction setting. As a result, I think this lack SOTA Imaging backbone and not trying different ways of extracting the imaging features, it may be possible that the limited contribution of imaging features is partially due to suboptimal feature extraction rather than an inherent lack of imaging signal. Could the authors please clarify whether any of the above experimental settings were tried?

2. Limited exploration of alternative multimodal fusion strategies - The multimodal setup relies primarily on feature concatenation followed by linear layers, which is very commonly applied in multi-modal settings. This late-fusion strategy may encourage the model to default to dominant clinical signals. Other fusion mechanisms (e.g., gated fusion, cross-modal attention, or modality dropout) could potentially force the network to leverage imaging features more effectively. Have the authors also tried these?

3. Feature attribution analysis for imaging is limited - The conclusion that imaging features are largely ignored is based mainly on the linear-layer weight magnitudes after feature concatenation and the calculated prediction overlap with logistic regression (78% of the test cases remained the same). While this is informative, these analyses do not directly attribute predictions back to specific image regions or image-derived features, making it difficult to determine where the imaging signal actually fails. Is there any other way such as GradCAMs that the authors did give a try?

4. Absence of imaging priors - Stroke outcome is strongly linked to infarct volume, core/penumbra mismatch, collateral circulation, and lesion location. The authors have relied on end-to-end feature learning without explicitly guiding the model toward these known prognostic factors, which may limit its ability to exploit imaging information.

**Detailed Comments:**

Please see below comments to further improve this work.

1. Explore more advanced image representations - To strengthen and also more effectively justify the claim that imaging provides limited added value, the authors should consider incorporating lesion-focused representations (e.g., infarct segmentation masks, infarct volume estimates, or regional lesion load), using SOTA imaging architectures such as DINOv3 and multi-scale or region-aware architectures that explicitly encode spatial context relevant to stroke outcomes, and evaluating whether performance changes when models are constrained or guided to focus on infarct-related regions (to name a few ways). Even a limited experiment or discussion acknowledging these alternatives would help clarify whether the negative result is architectural or indeed fundamental.

2. Investigate alternative multimodal fusion strategies - The study could benefit from testing or discussing fusion methods beyond simple concatenation, such as cross-attention between clinical tokens and imaging tokens, gated multimodal fusion that adaptively weights modalities per patient, modality dropout or regularization schemes that prevent the model from collapsing onto clinical features alone. This would help determine whether the observed dominance of clinical features is an artifact of the fusion design.

3. Use complementary feature attribution methods for imaging - The interpretability analysis could be strengthened by incorporating additional attribution techniques, such as GRADCAM on imaging inputs, occlusion or perturbation-based analyses (e.g., masking infarct regions or vascular territories), and comparing attribution consistency across correctly and incorrectly classified cases. These methods could provide spatial insight into whether the model truly ignores imaging or simply fails to extract clinically meaningful patterns.

**Justification Of Final Rating:**

The authors have addressed my comments in a reasonable manner. Although many experiments (such as including the segmentation, ROI guided imaging feature extraction) could not be tried due to the nature of the dataset, at least trying a pretrained large medical imaging 3D encoder has provided me much confidence into the current experiment evaluation from the imaging side. Moreover, the choice of simple concatenation for different modality features is also well justified as they want to keep this simple to make sure that the contribution of modalities is very well analyzed. Additionally, the interpretability analysis has been improved now in the revised manuscript.

Overall, I think this is a good systematically evaluated study on a large dataset to actually check the contribution of imaging features to the multimodal combination. Many works recently are simply combining the clinical and imaging features without actually systemtatically evaluating whether the imaging features could contribute and I think this will be a good paper for those studies to refer - both technical and clinical audience working in this community.

For the future extension, I would suggest the authors to go even more deeper. Imaging modality definitely should contain some useful features for making the final outcome prediction in my opinion? Then how these useful features could be explored and extracted in meaningful manner rather than the simple way of putting them through a image encoder and combining with the clinical factors. Does textual representations (of the imaging features) can play an improved role in this kind of combination? Or is there any other way that a specialized architecture for image extraction could be designed? I would really encourage the authors to make this extension and it would be even a more fantastic contribution, given that they have such a large dataset from various resources to validate these techniques.

**Justification Of The Preliminary Rating:**

This work is timely and important and is a valuable study that addresses a critical assumption in multimodal medical AI for stroke outcome prediction. However, clarification on the imaging and attribution methodologies should be expanded and strengthened to clearly validate the claims of the paper and then it would make a really strong contribution to the research community. With clearer positioning of the modeling choices relative to the SOTA and experimentation with alternative attribution and fusion strategies, the work would be even more compelling.

**Questions To Address In The Rebuttal:**

Please refer to the Weakness and Deatiled Comments section.

---

> ### Author Response · Authors · 2026-01-24
> **Response**
>
> Thank you for the detailed feedback. Please see the answers below:
>
> **Comment**: Explore more advanced image representations - To strengthen and also more effectively justify the claim that imaging provides limited added value, the authors should consider incorporating lesion-focused representations (e.g., infarct segmentation masks, infarct volume estimates, or regional lesion load), using SOTA imaging architectures such as DINOv3 and multi-scale or region-aware architectures that explicitly encode spatial context relevant to stroke outcomes, and evaluating whether performance changes when models are constrained or guided to focus on infarct-related regions (to name a few ways). Even a limited experiment or discussion acknowledging these alternatives would help clarify whether the negative result is architectural or indeed fundamental.
>
> **Response**: We agree that incorporating lesion information like masks and volume estimates should guide the model to learn better representations. However, our dataset only contained infarct volume estimates 24 hours after treatment, and it was not feasible to segment the entire dataset. In previous experiments, we tested how important the 24-hour infarct volume is and trained our clinical-only models using the baseline variables and the follow-up volume. This resulted in a 12% in AUROC and a 7% increase in AUPRC, with the shap analysis showing that the variable has the biggest impact on the model. This prompted us to pretrain the imaging backbones on predicting the follow-up volume from the baseline scans, then finetune on the 90-day mRS. However, the pretraining phase was not successful, and the model's loss remained constant despite trying different hyperparameter configurations. It is worth noting that predicting follow-up infarcts from baseline imaging is a challenging problem as shown by the recent ISLES24 results (https://arxiv.org/pdf/2408.10966). We updated our manuscript to discuss these limitations in our study.
>
> **Comment**: Investigate alternative multimodal fusion strategies - The study could benefit from testing or discussing fusion methods beyond simple concatenation, such as cross-attention between clinical tokens and imaging tokens, gated multimodal fusion that adaptively weights modalities per patient, modality dropout or regularization schemes that prevent the model from collapsing onto clinical features alone. This would help determine whether the observed dominance of clinical features is an artifact of the fusion design.
>
> **Response**: We completely agree with this comment. The main reason behind using simple concatenation is that 1) all literature we have seen except (https://www.sciencedirect.com/science/article/pii/S1361841524003062) uses this approach for feature fusion while reporting better performance than logistic regression, and 2) it makes it easier to investigate the weights following concatenation. Our goals were to investigate whether this fusion strategy that is used by almost all relevant studies leads to better performance than traditional methods like logistic regression and to validate whether the model learns anything useful from the image. We will include this limitation in our discussion.
>
> **Comment**: Use complementary feature attribution methods for imaging - The interpretability analysis could be strengthened by incorporating additional attribution techniques, such as GRADCAM on imaging inputs, occlusion or perturbation-based analyses (e.g., masking infarct regions or vascular territories), and comparing attribution consistency across correctly and incorrectly classified cases. These methods could provide spatial insight into whether the model truly ignores imaging or simply fails to extract clinically meaningful patterns.
>
> **Response**: We fully agree with that comment. We performed Shap analysis on the deep learning models and found that 1) the input clinical features have a bigger contribution to the prediction and 2) the outer edges of the brain have large Shap values compared to the rest of the brain. We included our findings in the updated manuscript.

---

> > ### Comment · Reviewer_kpmd · 2026-01-25
> >
> > Thanks to the authors for updating the manuscript and responses to the comments. I appreciate it. Could you please clarify the below points
> >
> > 1. Explore more advanced image representations  - Did the authors try any pretrained image encoder backbone - for e.g., DINOv3 or something that has been trained on large scale medical database and images - such as BiomedGPT or something else as the feature encoder? I understand that incorporating the segmentation lesion masks and ROI will be challenging as the dataset only contains infarct volume estimates 24 hours after treatment. However, extracting the image features from large scale encoders should still be achievable. I ask this because a lot depends on how good the extracted representations are, which eventually decides their contribution to the final decision making.

---

> > > ### Author Response · Authors · 2026-01-26
> > > **Response**
> > >
> > > We tried the Med3D pretrained image encoder backbone (https://arxiv.org/pdf/1904.00625) when training the NCCT + Clinical model. This backbone was originally trained on 8 datasets containing segmentations of different tumor types and organs. In a previous study that we have cited in our manuscript (https://doi.org/10.3389/FNEUR.2022.809343), they used the pretrained Med3D backbone to train on CTA and clinical features. While their results improved substantially when using it, they 1) only reported cross-validation results, and 2) the logistic regression model had **4%**  better AUROC.
> > >
> > > In experiments that we have not included in the manuscript, we trained the NCCT + Clinical model with the Med3D backbone and found that 1) it had similar performance (AUROC=0.71, AUPRC=0.83) to SEResNext and TranSOP, and 2) the weight heatmap showed a similar pattern to the one we show in the manuscript (i.e., larger weights for the clinical features).

---

> > > > ### Comment · Reviewer_kpmd · 2026-01-26
> > > >
> > > > Thank you for that clarification and more information on the experiments tried with a pre-trained medical image 3D backbone encoder.
> > > >
> > > > It's really interesting to see that compared to SEResNext and TranSOP, the performance almost remains the same, and thus I think the weighted heatmap would have larger weights for the clinical features.
> > > >
> > > > Can I please suggest that you include this experiment in the Appendix of the manuscript, if not in the main paper, that's okay. But the authors should definitely mention that a large pretrained 3D imaging model was at least tried (albeit a little old one) and tested. This provides confidence into the the extracted features from the imaging modality.

---

> > > > > ### Author Response · Authors · 2026-01-26
> > > > > **Response**
> > > > >
> > > > > We will include the experiment in the final version of the manuscript. As you mentioned, the next logical step for this study is to train an image encoder that learns better image representations. We believe imaging contains useful information, as demonstrated by our experiment using the 24-hour infarct volume.
> > > > >
> > > > > Lastly, thank you for your valuable feedback. It helped improve the manuscript.

---

### Official Review · Reviewer_TS1y · 2026-01-16

**Confidence:** 4
**Preliminary Rating:** 4
**Final Rating:** 4

**Summary:**

The paper “Validating the Benefit of Combining Imaging and Clinical Data for Ischemic Stroke Outcome Prediction” focuses on predicting patient stroke outcomes using baseline imaging and clinical data. This is an important problem, as acute stroke is a leading cause of disability and death, and faster, more accurate decision making can lead to improved patient outcomes. Multiple studies have used simple logistic regression on clinical variables to estimate the mRS. While these features are informative, they do not capture the full imaging information present in medical scans. However, when machine learning methods are trained solely on medical images they often underperform compared to traditional clinical models. It is only when clinical variables and imaging data are combined that these approaches begin to match or slightly outperform the performance of traditional methods.

For this reason, the authors study the effect of combining these two modalities. They first train models using either clinical data or imaging data alone, and then analyze the model weights to better understand which modality has a greater impact on the predictions. Finally, they investigate whether the imaging derived features meaningfully contribute to outcome prediction.

**Strengths:**

The paper addresses a highly relevant and clinically meaningful problem, namely the prediction of functional outcome in acute stroke, which remains challenging even for clinicians. The authors provide a concise and well structured overview of prior work, clearly identifying gaps and ongoing difficulties in effectively integrating imaging and clinical information for outcome prediction.

A major strength of the study is the large, multi-center dataset comprising over 1,100 patients, which enables a robust evaluation and increases the robustness of the findings. The experimental design is thorough, with the training of multiple models across different modality combinations, allowing for a thorough assessment of the contribution of clinical and imaging features. In particular, the analysis of clipped model weights and the ablation of highlighting vascular versus infarct related image regions for CTA imaging were interesting.

Overall, the paper is clearly written and well structured, with a concise analysis that presents several insightful findings aimed at understanding why approaches that combine imaging features may underperform.

**Weaknesses:**

One limitation of the study relates to the dataset splitting strategy. The authors group all samples by care center and assign each center exclusively to either the training or test set. While this approach is intended to mitigate center specific bias (as stated), it also introduces a strong inter center domain shift that may impact the model performance. This could be relevant because, as noted in the paper, the trial investigators did not enforce a standardized imaging acquisition protocol, and each center was allowed to use its local scanning procedures. These acquisition differences could potentially lead to systematic distribution differences between centers. The chosen split strategy may therefore negatively affect image based models. Additional experiments, such as within-center validation would help clarify the impact of this choice.

**Detailed Comments:**

A very minor point is that NCCT imaging is not explicitly defined in the paper (though presumably refers to non-contrast CT)

**Justification Of Final Rating:**

I think this is a good paper that evaluates the impact of imaging features in a multimodal setting. Many works combine clinical and imaging features without studying the impact each modality has on the final outcome. This paper provides a useful analysis for this specific dataset, which helps inform future work on this problem. Particularly, it would be interesting to explore why these imaging features, despite containing useful information, do not appear to contribute much to the final prediction. Moreover, this shows that simply merging modalities and having more data does not necessarily lead to improved performance and that the contribution of each modality should be better understood.

Furthermore, the authors have addressed all my comments and added the requested experiments in the appendix, which improve the interpretability analysis and clarify the impact of the dataset in the revised manuscript.

**Justification Of The Preliminary Rating:**

Weak accept. The paper addresses a clinically important and timely problem and provides a careful, well structured validation of multimodal stroke outcome prediction models. The study offers insight into why combining imaging and clinical data yields limited performance gains in practice. This was supported by SHAP analysis to identify influential clinical features and weight heatmap visualizations for the combined models. Furthermore, the use of a large, multi-center dataset strengthens the robustness of the analysis, and the vascular versus infarct ablation is an interesting finding. While the center based data split, complicate the interpretation of the results, the paper nevertheless makes a strong contribution.

**Questions To Address In The Rebuttal:**

In the discussion, it is mentioned that adjusting the CTA window to better highlight infarcted regions substantially improves performance, nearly matching that of NCCT. When combining all modalities, was CTA also evaluated using this alternative windowing strategy? The multimodal model currently achieves an AUROC of approximately 0.66, and it is unclear whether suboptimal CTA preprocessing may have limited the combined performance.

The authors group samples by care center and assign each center exclusively to either the training or test set in order to reduce potential bias. Could the authors clarify the rationale for this choice, and whether its impact on model performance was explored? This could be relevant because, as noted in the paper, the trial investigators did not enforce a standardized imaging acquisition protocol and each center was allowed to use its local scanning procedures.

---

> ### Author Response · Authors · 2026-01-24
> **Response**
>
> Thank you for the detailed feedback. Please see the answers below:
>
> **Comment**: A very minor point is that NCCT imaging is not explicitly defined in the paper (though presumably refers to non-contrast CT)
>
> **Response**: We updated the manuscript in section 2.1 to explicitly define NCCT as non-contrast CT
>
> **Comment**: In the discussion, it is mentioned that adjusting the CTA window to better highlight infarcted regions substantially improves performance, nearly matching that of NCCT. When combining all modalities, was CTA also evaluated using this alternative windowing strategy? The multimodal model currently achieves an AUROC of approximately 0.66, and it is unclear whether suboptimal CTA preprocessing may have limited the combined performance.
>
> **Response**:  The suboptimal CTA preprocessing reduced the performance when combining all modalities. When adjusting the CTA window to highlight infarcts, AUROC increased by 3%. We updated the manuscript to include this result.
>
> **Comment**: The authors group samples by care center and assign each center exclusively to either the training or test set to reduce potential bias. Could the authors clarify the rationale for this choice, and whether its impact on model performance was explored? This could be relevant because, as noted in the paper, the trial investigators did not enforce a standardized imaging acquisition protocol, and each center was allowed to use its local scanning procedures.
>
> **Response**: We wanted an independent test set from different centers to prevent any site or imaging acquisition information from correlating with the target variable which can result in inflated test performance. To investigate the effect of distribution shift between centers, we pooled all patients from all centers and split randomly. In the new splits, different patients from the same centers appeared in the training and test sets. NCCT-only improved AUROC and AUPRC by 4%, and 3%, respectively. NCCT+Clinical dropped by 1% for AUROC and AUPRC. However, the same pattern persists in the weights' heatmap. When investigating the weights for the NCCT+Clinical model, the weights associated with the clinical variables still dominated. This indicates that even though the imaging only models improved slightly with the new splits, the combined model mainly focused on the clinical ones.  We included a table of the results in the appendix.

---

> > ### Comment · Reviewer_TS1y · 2026-01-26
> > **Response**
> >
> > Thank you to the authors for updating the manuscript and for the responses to the comments. In particular, thank you for including the additional experiment suggested in the appendix.
> >
> > These results (61% AUROC) further suggest that the representations extracted by the imaging backbone may not be expressive enough to capture the relevant features needed to reliably predict the outcome. Furthermore, I agree with the discussion’s conclusion that improving these learned representations will be key to enhancing the performance of both the imaging-only and the combined imaging-plus-clinical models and is a good next step.

---

### Author Rebuttal · Authors · 2026-01-24

**Rebuttal:**

We thank the reviewers for the detailed and constructive feedback on how to improve this paper. We have addressed all their comments and attached a revised manuscript. The revised manuscript improves on our analysis and includes explainability maps that highlight brain regions of interest to the models. This analysis supports our preliminary findings that the models mainly focus on clinical data. Furthermore, we included additional experiments in the appendix that investigate the effect of our splitting strategy.

**Update**: We uploaded the revised manuscript following the reviewers' feedback during the rebuttal and discussion phases. We thank all reviewers for their valuable input.

**Supporting Material:**

/attachment/064e5bd51d498ca1e638b31b220f83fb16936a34.pdf

---

### Meta-Review · Area_Chair_yhMz · 2026-02-05

**Recommendation:** Accept (Poster)
**Confidence:** 5

**Metareview:**

All reviewers agree that the paper brings moderate technical innovation but has a definite and timely methodological importance. They praise the authors for their rigorous and rich experimental evaluation. The authors were willing and reactive to address the questions and remarks of the reviewers. Using more diverse and complete image data and annotations, as well as more advanced image representations was recognized as a path to more definitely confirm, or mitigate, the authors' conclusion that image data has very limited impact on ischemic stroke outcome prediction.

---

### Decision · Program_Chairs · 2026-02-13

Accept (Poster)